# Investigation on the Preparation and Properties of CMC/magadiite Nacre-Like Nanocomposite Films

**DOI:** 10.3390/polym11091378

**Published:** 2019-08-22

**Authors:** Mingliang Ge, Yueying Li, Yinye Yang, Yanwu Wang, Guodong Liang, Guoqing Hu, Jahangir Alam S.M.

**Affiliations:** 1Key Laboratory of Polymer Processing Engineering of Ministry of Education, National Engineering Research Center of Novel Equipment for Polymer Processing, School of Mechanical & Automotive Engineering, South China University of Technology, Guangzhou 510640, China; 2School of Material Science and Engineering, Guizhou Minzu University, Guiyang 550000, China; 3Key Laboratory of Polymeric Composite & Functional Materials of Ministry of Education, Sun Yat-Sen University, Guangzhou 510275, China; 4School of Engineering, The University of British Columbia, Kelowna V1V 1V7, Canada; 5Dept. of Robotics & Mechatronics Engineering, University of Dhaka, Dhaka 1000, Bangladesh; 6Faculty of Engineering & Technology, Jashore University of Science and Technology, Jessore 7408, Bangladesh

**Keywords:** sodium carboxymethylcellulose, magadiite, nacre-like, nanocomposite film

## Abstract

The layered hydrated sodium salt-magadiite (MAG), which has special interpenetrating petals structure, was used as a functional filler to slowly self-assemble with sodium carboxy-methylcellulose (CMC), in order to prepare nacre-like nanocomposite film by solvent evaporation method. The structure of prepared nacre-like nanocomposite film was characterized by Scanning Electron Microscope (SEM) and X-ray diffraction (XRD) analysis; whereas, it was indicated that CMC macromolecules were inserted between the layers of MAG to increase the layer spacing of MAG by forming an interpenetrating petals structure; in the meantime, the addition of MAG improved the thermal stability of CMC. The tensile strength of CMC/MAG was significantly improved compared with pure CMC. The tensile strength of CMC/MAG reached the maximum value at 1.71 MPa when the MAG content was 20%, to maintaining high transparency. Due to the high content of inorganic filler, the flame retarding performance and the thermal stability were also brilliant; hence, the great biocompatibility and excellent mechanical properties of the bionic nanocomposite films with the unique interpenetrating petals structure provided a great probability for these original composites to be widely applied in material research, such as tissue engineering in biomedical research.

## 1. Introduction

Researchers reported that many naturally-occurring materials, such as the wings of insects or the beaks of woodpeckers, have unique structures, which endow them with specific properties [1]. Their structures often provide inspirations for researchers to design materials that meet particular functions [2,3]. Biomimetic materials had been architected by simulating the structural characteristics and biological functions of natural substances at the molecular level, and then the scientists could develop new materials that resemble or even surpass the functions of the original natural substances [4,5]. One such example had been found in nacre, a common natural organic-inorganic composite material [6,7].

Though composed of relatively soft components (95% brittle aragonite tablets and 5% organic matter (mainly β-chitin and silk fibroin)), it had possessed intriguing mechanical properties that exhibit exceptional damage tolerance resulting from their precise hierarchical structures over several length scales, deriving from nanoscale building blocks [8,9]. The Mollusca had evolved efficient strategies to synthesize order layer-shaped “brick-motar” (a “motar” being a fabricator of brick) characteristic microstructures, which is similar to modern architecture, to prepare a tough model structure of nacre [10,11]. It is the complex multi-stage structure that makes the nacre both high in strength and toughness. This combination of high strength and high toughness was usually not common with artificial materials [12]. Inspired by nacre’s structure, scientists used the functional inorganic layered material of a nanoscale or microscale two-dimensional structure to play the role of “brick”; in the meantime, organic polymer was chosen as the "motar" to prepare this imitation shell “brick” [13].

Up to now, scientists have synthesized a large number of nacre-like materials with impressive mechanical properties, which were based upon different types of material and manufacturing processes. In recent years, different methods such Layer-by-layer (LBL), ice-templating and sintering (frottage) have been applied for the fast fabrication of nacre-like films. For example, Yao et al. [14] prepared a series of layered, organic-inorganic hybrid films, reinforced with layered double hydroxide (LDH) micro-and nano-platelets through an LBL assembly procedure, using a series of LDH platelets as building blocks. Walther et al. [15] prepared a novel nacre-mimetic paper composites via a paper-making process, which was similar to the starting process; recently, Woo’s group [16] reported that they prepared a graphene oxide/cross-linking agent (GO/CA) composite inspired by the nacre structure, in which interfacial engineering was applied. Vacuum filtration is also an important method for preparing nacre-like film. For instance, Huang et al. [17] prepared nacre-like polydopamine-coated clay (D-clay) films, utilizing polydopamine (PDA) and clay nano-sheets by the vacuum filtration-assisted assembly method. Immersed in an aqueous solution of Fe^3+^ ions, the obtained D-clay/Fe^3+^ films had significantly enhanced mechanical properties; and compared to above mentioned methods, the solution-casting method was a simple, time-saving, and easy to scale up method for nacre-like structures. For instance, Li et al. [18] prepared R–PVA/GO composite films with good biocompatibility, superior mechanical properties and high electrical conductivity by a simple solution-casting method followed by a post-reduction treatment procedure.

A number of synthetic and natural polymers, such as polyethylene glycol, polyvinyl alcohol (PVA) [17], poly(methyl methacrylate) [18], chitosan [19], and silk fibroin [20] had been utilized as adhesive and soft components which could be described as a “motar” in nacre. Functional inorganic sheet materials, including zirconium phosphate [21], graphene [22,23,24], alumina [10], silver nanoparticles [25], layered double hydroxide (LDH) [26], hydroxyapatite [27], layered silicate [28,29] etc. had usually been taken as “brick” in the fabrication of nacre-like hybrid film. In recent years, many novel inorganic materials had been selected as hard phases. It had been reported that Zhao [30] assembled a nacre-mimetic graphene/poly(vinyl alcohol) nanocomposite film with both asperities and bridges introduced to the layers in which graphene had chosen as the brick. Mäkiniemi et al. [31] reported in detail how to integrate attractive electric and ionic conductivity into mechanically robust self-assembled nacre-like material using highly functional polymers (PEDOT:PSS and PILs) in combination with a synthetic saponite nano-clay (SUM). Liang’s group [19] selected MMT as “brick” successfully, and fabricated strong and tough CMMT–ALG nanocomposites though introducing the cross-linking of alginate with Ca^2+^ into the field of artificial nacre by the method of the vacuum-assisted filtration self-assembly process (VAF). However, nacre-like films had been prepared by other groups that had been reported using the materials with parallel layers to act as “brick” phases [32]. The parallel arrangement of lamellar structure had been determined that it was prone to slippage between the sheets. The existence of hydrogen bond attraction forces were unstable. The research said that a special force of MAG between the layers, such as the blocking effects, would manage this problem very well [33,34].

It had been reported that carboxymethylcellulose (CMC) was a derivative of cellulose, consisting of carboxy-methyl groups attached to the polysaccharide backbone, and was the widest used natural polysaccharide polymer, with good biodegradability, biocompatibility and film formation [35,36]. Due to its safety and being non-toxic, it had been widely used in the chemical, food and packaging industries [37]. As a hydrophilic polysaccharide polymer, the film forming quality of CMC was relatively poor and the film was brittle, had poor moisture resistance, and was generally terrible in thermal stability [38]. Previous work had shown that the exfoliation and dispersion of nano-clay particles in the nano-cellulose matrix had a large influence on the mechanical properties of the nano-cellulose–nano-clay composites [39,40]. In order to improve the performance of CMC, many researches had been done in this domain. For instance, Das [41] et al. demonstrated the preparation of thickness, self-assembling, hybrid fire barrier coatings of sodium carboxy-methyl cellulose (CMC)/montmorillonite (MTM) with well-defined, bioinspired brick-wall nanostructure through a single-step, water-borne approach; whereas, it had improved the fire barrier and retardancy properties of CMC.

MAG with good adsorption and ion-exchange properties had been chosen as the “brick” in this paper; at a first time, MAG was found in the soda-lakes in Kenya [42], and it had been reported that MAG is simply and conveniently synthesized in the laboratory by a hydrothermal synthesis method nowadays [43,44]. In comparison to other common layered silicate, apart from this, MAG has layers that had certain radian, forming interpenetrating petals structure when they aggregate [45]; and this goes for the sodium carboxymethylcellulose (CMC), which has great biocompatibility to be used as the “motar”, to fabricate a novel nacre-like nanocomposite film by the casting "solvent evaporation" method. The correlational researches about nacre-like CMC/MAG nanocomposite film have not been reported at home and abroad yet. The structure, thermal stability, mechanical performance, transparency and flame retarding properties of the CMC/MAG nanocomposite film was formed by different contents of MAG which were compared under the same conditions. The results indicated that the mechanical behavior of the nacre-like nanocomposite film was effectively improved, especially tensile strength. Based upon the thermal gravimetric (TG) analysis and combustion tests, the thermal stability and flame resistance of CMC/MAG composite films were enhanced considerably, as well as maintaining a certain degree of transparency; moreover, there has remained some suspense for further explore all of the other outstanding properties which have not yet been investigated deeply in other researches.

## 2. Materials and Methods

### 2.1. Materials

The magadiite (MAG) was prepared using a hydrothermal synthesis method in the laboratory [44,46]. The carboxymethylcellulose (CMC) was purchased from the Pharmaceutical Group Co., Ltd. (Guangzhou, China). Other chemical reagents such as HCl and NaOH were purchased from Guangzhou Qianhui Company (Guangzhou, China).

### 2.2. Preparation of CMC/MAG Nacre-Like Nanocomposite Films

MAG and CMC were dried at 50 °C for 10 h. According to the mass ratio of MAG: CMC: H_2_O: 10:90:1000, 20:80:1000, 30:70:1000, 40:60:1000 and 50:50:1000, the raw materials were weighed, respectively. The MAG and water were placed in a blade stirrer for 10 min, then added to CMC and the mixture were stirred for 5 min; the mixture was placed in a watch glass, then it was dried in an oven at 50 °C. The obtained product was recorded as CMC/MAG-*n* (*n* is the content of MAG in %).

### 2.3. Characterization

X-ray diffraction (XRD, D8 ADVANCE, Bruker AXS, Karlsruhe, BW, Germany) analysis was recorded on a D8 Advance X-ray diffractometer with Cu Kα radiation (λ= 1.5406 Å). The morphology of the sample was observed using a Leo 1530 vp scanning electron microscope (SEM, Zeiss, Oberkochen, BW, Germany).

The sample was adhered to the test bench with conductive adhesive for gold spraying; the operating voltage range was 10–20 kV. The thermal stability of the membranes was examined using thermal gravimetric analysis (TGA, STA449 C, NETZSCH, Selb, Bavaria, Germany) and differential thermal gravimetric (DTG). The tensile properties of materials were tested on an Instron 5566 universal testing machine (Intron, Boston, MA, USA). The tensile rate is 2.00 mm/min, the spline standard is 10×80 mm, and the film thickness is 0.50 to 0.80 mm.

## 3. Results and Discussion

### 3.1. XRD Analysis

Figure 1 shows the XRD spectrum of the CMC/MAG–20 and CMC/MAG–40 respectively. It can be seen that CMC/MAG–20 has a large number of different intensities peaks at 2*θ* of 1.15°–2.00°, forming an apparent platform. The corresponding layer spacing is between 4.40 nm and 7.66 nm according to the Bragg equation; and this was due to the fact that the CMC macromolecules, whose volume was much larger than the original MAG interlayer space, entered the MAG layers, and the MAG’s layer spacing was enlarged by the volume effect. However, when the content of MAG was added to 40%, the counterpart on XRD did not exhibit a same platform, but was similar to that of the pure MAG. It was because of the fact that in the case of excessive MAG content, it was difficult for CMC macromolecules to enter the interlayer of MAG, which is closely related to the aggregation of MAG which happened in the process of dispersion. The 001-diffraction peak of MAG is at 2*θ* = 5.77°, and the corresponding layer spacing was 1.53 nm. At the same time it can be seen from the Figure 1 that CMC/MAG–20 and CMC/MAG–40 both have a 001-plane diffraction peak at 2*θ* = 5.80°, which was with little difference compared to pure MAG. These consequences indicate that some of the MAG has not been entered by the CMC macromolecular chain, resulting in the XRD-exhibited characteristic peak of MAG. The reason for not entering the CMC macromolecular chain was that on the one hand, the amount of MAG is large, it failed to fully contact with CMC molecules, so that the CMC macromolecular chains did not enter. On the other hand, due to agglomeration of MAG, some of them remained as relatively large constrictive spheres rather than becoming uniformly dispersed, leading to not being entered by the CMC macromolecular chain during the mixing process and "solvent evaporation". The CMC/MAG–20 and CMC/MAG–40 XRD spectrum both indicate that the CMC was successfully composited with MAG, however, the different amount of MAG led to different structure, which distinguished their performance in the following investigation.

### 3.2. SEM Analysis

Figure 2a,b show the SEM images of CMC/MAG–20 and CMC/MAG–40, respectively. It can be seen from the figures that most of the MAG rose petal-like microspheres in CMC/MAG–20 were peeled off; the layer spacing was stretched up to varying degrees (shown in Figure 2b II), and a part of the MAG exists in a monolayer structure (as shown in Figure 2a I), indicating that the part of the MAG has been completely peeled by the CMC macromolecule. The CMC macromolecule chain was combined with the MAG layer through the hydrogen bond which was formed between –OH, –COOH; however, –H on the CMC and –OH on MAG layer (as shown in Figure 2a I). But there still existed some conglobate MAG, the depression zone (Figure 2b III), which could be inferred that it was formed by the fall of spherical MAG, which can prove that there was an aggregating spherical MAG particle. The structure of rose petal-like microspheres in CMC/MAG–40 was seen nearly unbroken (Figure 2c IV). There was flocculent structures between the layers of most of the rose petal-like microspheres and rose petal-like microspheres were stretched in a relatively small extent (Figure 2d V). In general, CMC/MAG–20 exhibited lamellar separation and a nearly parallel structure while CMC/MAG–40 preserved the rose petal-like morphology similar to MAG. It was found that CMC/MAG–40 has not a separate CMC matrix compared to CMC/MAG–20, which indicated that 60% of the CMC macromolecules stayed outside of the MAG layers; moreover, CMC/MAG–40 was basically formed by rose petal-like microspheres. These results corresponded to the XRD analysis.

The layer of MAG was not completely straight like a familiar montmorillonite lamella, but had a certain curvature which was beneficial to the blocking effect-lamellas were not prone to relative slippage when being stretched, as shown in Figure 3. The original structure of MAG consisted of tight rose petal-like microspheres in which curve layers formed spheres from the inside out. Once the moderate ratio of MAG was composited with CMC, the constrictive layers were separated by long CMC chains; whereas, it was broken in its spherical structure and the interlayer space than that it was enlarged. The inserting of CMC could have improved the tensile property of CMC/MAG for at least two reasons. For one thing, the CMC chains between layers increased adhesion between layers; and for another reason, it was seen for distinguishing CMC/MAG from other composites; therefore, the curve layers were like a tenon structure, the protuberances of the upper layer, and the depressions of the lower layer lock into each other; thus, that they were hardly prone to slippage along the direction of the force.

### 3.3. Thermo-Gravimetric Analysis

The CMC/MAG composite film has three losing weight areas in the range of temperatures 30 °C–250 °C; the weight loss area of adsorption water, combined water and the weight loss rate were about 16%, due to the film’s need to contain certain moisture during the preparation process, otherwise the surface would show serious wrinkles and irregularities; 250–340 °C was the fracture degradation process of the CMC molecular chain area, which is the main area of weight loss; whereas, over 340 °C was mainly the loss of carbon-containing substances. Figure 4 shows that the temperature at which degradation begins of CMC/MAG–20 and CMC/MAG–40 was significantly behind the CMC; more specifically, the degradation temperature of CMC was improved by 30 °C. It can be inferred that the addition of MAG significantly increases the degradation temperature of CMC, indicating that the addition of MAG improves the thermal stability of CMC; however, the higher the MAG content is, the better thermal stability the composite film would have. On the other hand, due to the fact that the CMC was surrounded by the MAG sheet with an “interpenetrating petals” layered structure, CMC molecular chains were intercalated into the interlayers of MAG. The thermal degradation temperature of CMC was raised to a higher degree resulting from MAG layers possessing excellent thermal stability and barrier properties. On the other hand, the hydrogen bonds and electrostatic force between the MAG layers and CMC molecular chains could enhance the thermal stability of the nacre-like film.

### 3.4. Mechanical Performance Analysis

Figure 5 shows the mechanical properties of the composite film. The tensile rate was 2.00 mm/min, the spline standard was 10 × 80 mm, and the film thickness is 0.50 to 0.80 mm. It can be seen from Figure 5, the tensile strength of the CMC–MAG film was significantly increased. When the MAG content was 20%, the tensile strength reached a maximum value at 1.71 MPa. When the MAG content is 30% and 40% or more, the tensile strength was not decreased so much. As the MAG content was increased, the material became brittle, and the rigidity increased greatly. The CMC/MAG nacre-like film has good tensile strength to the unique microscopic construction of MAG. Interpenetrating petals layered the structure of the MAG, which was the main reason for improving the mechanical properties of the nacre-like hybrid membranes. When the plastic deformation occurs to the film, a great number of chemical bonds and hydrogen bonds would be destroyed and it would be restructured; whereas, the result of the effect of the cementing action of the organic substrate improved the slip resistance, effective maintaining the interpenetrating petals layered structure inside the hybrid membranes, and delaying the fracture time, which explain the increase of tensile strength and elongation at the break point. When the content of MAG was at an appropriate amount (20%), CMC macromolecular between the layers stretched most of the MAG’s layers, and increased the slip resistance of layers. When the amount of MAG was added, however, more reunions of MAG microspheres took place. The agglomerated particles were distributed in the system by reducing the bonding force between the layers, but resulting in a decrease in tensile strength.

### 3.5. Transparency Analysis

Figure 6 shows the optical transparency of CMC and CMC/MAG nacre-like films, which was seen from Figure 6a,b compared with pure CMC and 10% content of MAG composite films’ transparency reduced rarely. From Figure 6c, CMC/MAG–20 have slight declined transmittance, but CMC/MAG–20 was maintained at a high light transmittance, and it was the relatively lower transmittance of CMC/MAG–20 which was associated with the microstructure of two of MAG. CMC/MAG was formed by the accumulation of rose petal-like microspheres composed of lamellar structure and the microspheres making the layers have no directionality. From Figure 6d, when the addition amount of MAG was increased to 40%, the patterns visible through the membrane were relatively fuzzy, and the transmittance was decreased. In terms of the overall color of the membrane, CMC/MAG–10 was slightly unclear; when the addition amount was 40%, CMC/MAG were obviously white, which was related to the color of MAG—the synthetic MAG was a pure white powder.

### 3.6. Combustion Analysis

Figure 7 shows CMC, CMC/MAG–20 and CMC/MAG–40 combustion diagrams. Exposed to the flame, the pure CMC film was frizzled immediately and burned very-rapidly with a large yellow flame (Figure 7a). From Figure 7b,c, it can be seen that CMC/MAG–20 and CMC/MAG–40 have fewer combustion flames and better flame retarding properties, but CMC/MAG burnt slower than pure CMC; however, they began with the surface hybrid films gradually darkening, mainly resulting from the carbonization of the CMC molecules. This is determined by the interpenetrating petals-layered structure of the MAG, protecting the CMC molecules surrounded in it. Due to the high proportion of inorganic fillers, the combustion process had hindered the contact of combustibles with oxygen. The CMC/MAG was composed of interpenetrating petals structure microspheres composed of lamellar structures which are non-directional and could form lots of dense fire-retardant layers, as well as discouraging further diffusion of the flame. With the increase of inorganic filler content from the comparison of the pictures in Figure 7b–d, the combustion ash became darker. As shown in Figure 7d, the scattered holes were found in the middle part after the combustion of CMC/MAG–40 which also was confirmed from another angle that a mass aggregation of MAG existed in it, so that the aggregate particles fell off from the film during burning.

## 4. Conclusions

In this research paper, MAG was used as functional filler to evaporate the solvent by casting method to slowly self-assemble with sodium carboxy-methyl cellulose (CMC). The CMC/MAG nacre-like nanocomposite film was prepared with different contents of MAG. The SEM and XRD analysis showed that CMC/MAG was formed by stacking rose petal-like microspheres composed of lamellar structures. The accumulation of rose petal-like microspheres made the structure non-directional, and the entry of CMC macromolecules between them increased the layer spacing. In the meantime, the adding of MAG improved the thermal stability of the CMC. The tensile strength of CMC/MAG was significantly improved compared with pure CMC. When the MAG content was 20%, the tensile strength of CMC/MAG reached the maximum value. Transparency analyses showed that the transparency of MAG was very high when the content was up to 20%. Due to the high content of inorganic filter, the flame retarding performance was good. Therefore, our designed multiscale assembly is an alternative approach to prepare strong functional composite materials, and the nacre-like composite membranes are very promising in the construction of highly tough and strong composites to be applied in biomaterial research, such as tissue engineering in biomedical research.

## Figures and Tables

**Figure 1 polymers-11-01378-f001:**
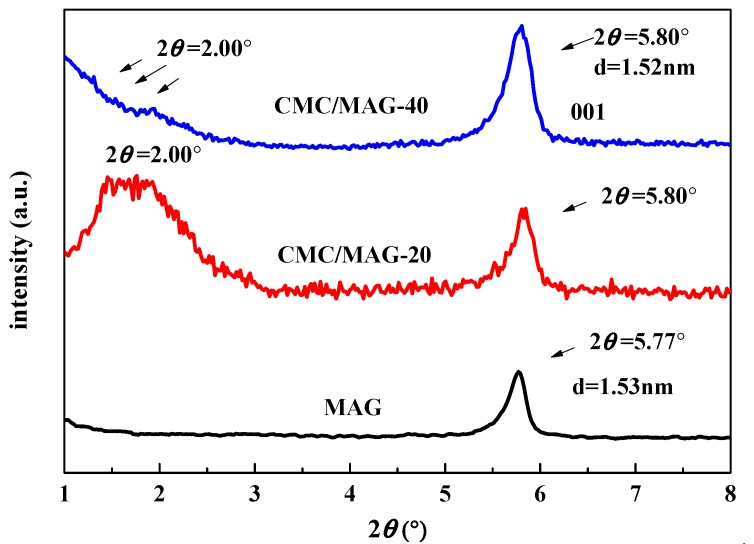
X-ray diffraction (XRD) patterns for magadiite (MAG) (curve black), carboxymethylcellulose (CMC)/MAG–20 (curve red) and CMC/MAG–40 (curve blue).

**Figure 2 polymers-11-01378-f002:**
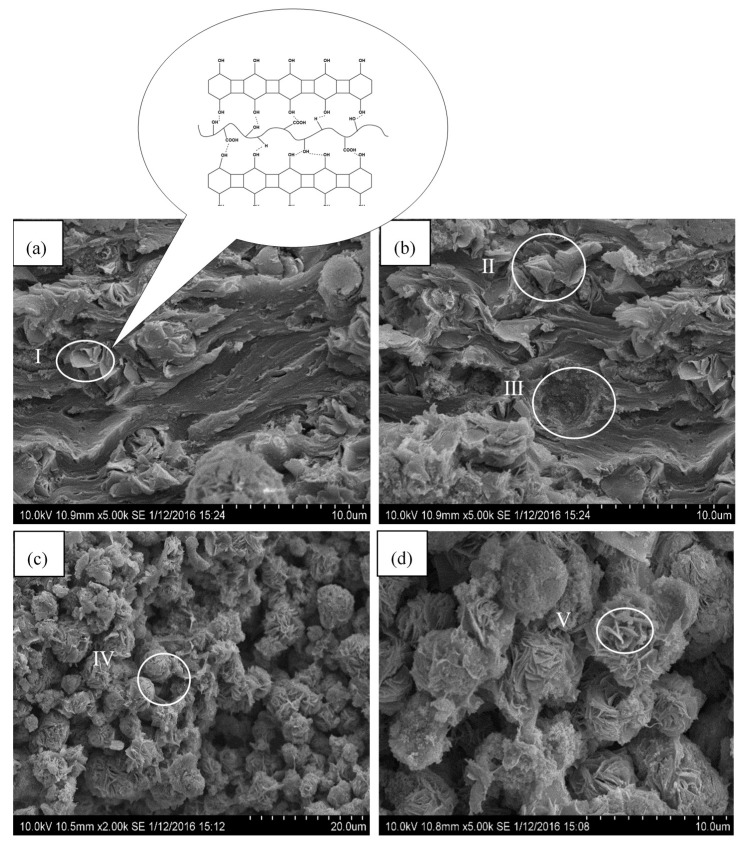
Scanning electron microscope (SEM) image of cutting fracture section of (**a**,**b**) CMC/MAG–20 (**c**,**d**) CMC/MAG–40.

**Figure 3 polymers-11-01378-f003:**
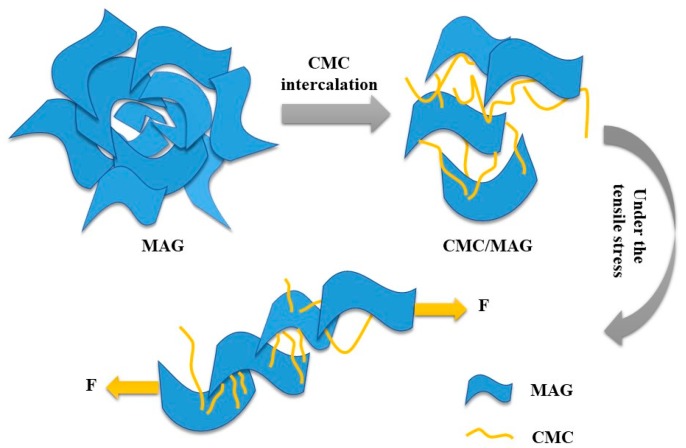
Schematic diagram of microstructure of CMC/MAG.

**Figure 4 polymers-11-01378-f004:**
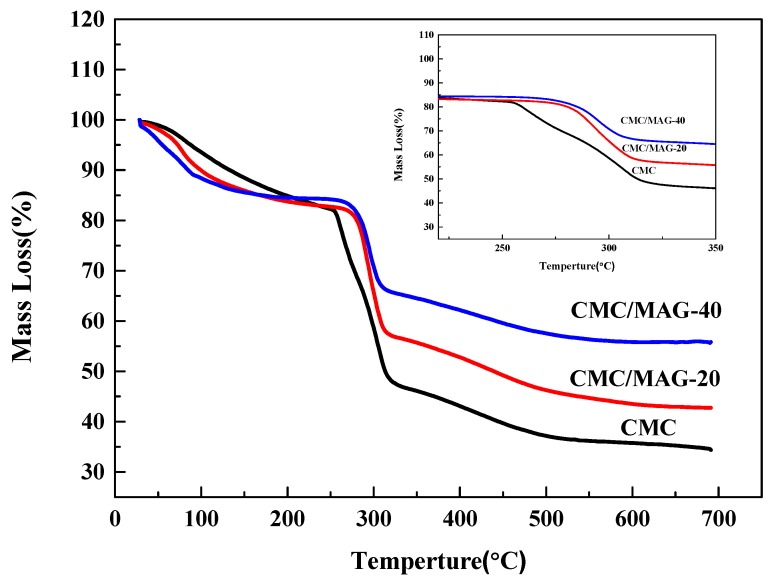
TG curves of CMC (curve black), CMC/MAG–20 (curve red) and CMC/MAG–40 (curve blue).

**Figure 5 polymers-11-01378-f005:**
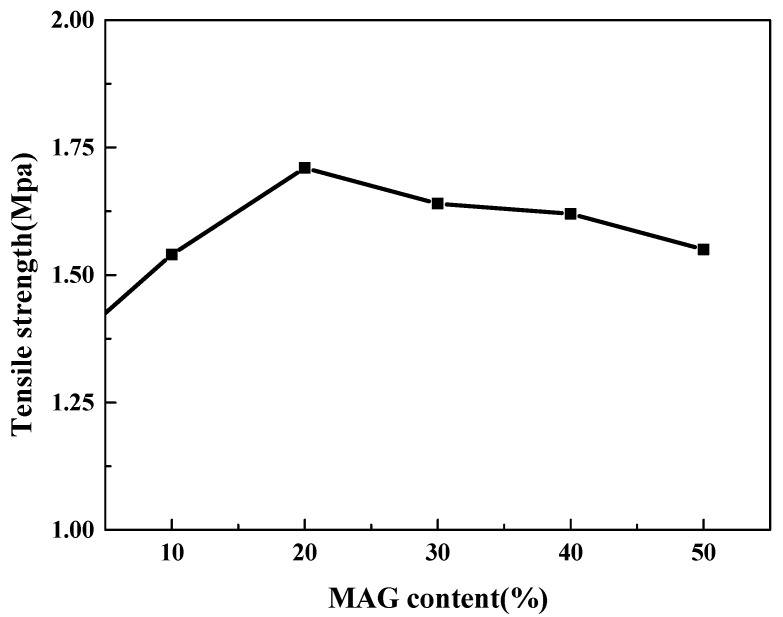
Mechanical properties of composite films.

**Figure 6 polymers-11-01378-f006:**
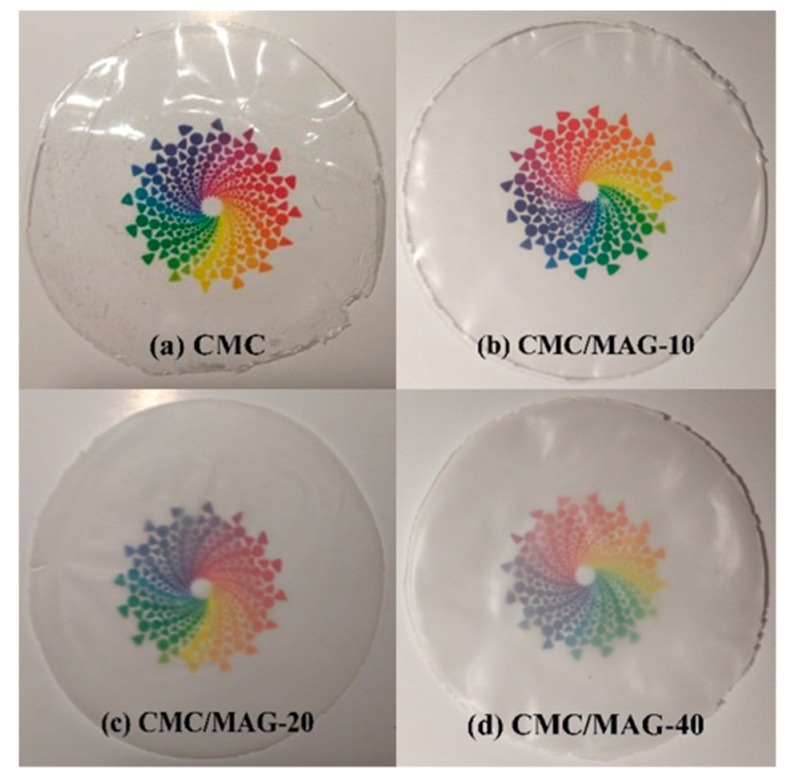
Optical transparency of CMC and CMC/MAG.

**Figure 7 polymers-11-01378-f007:**
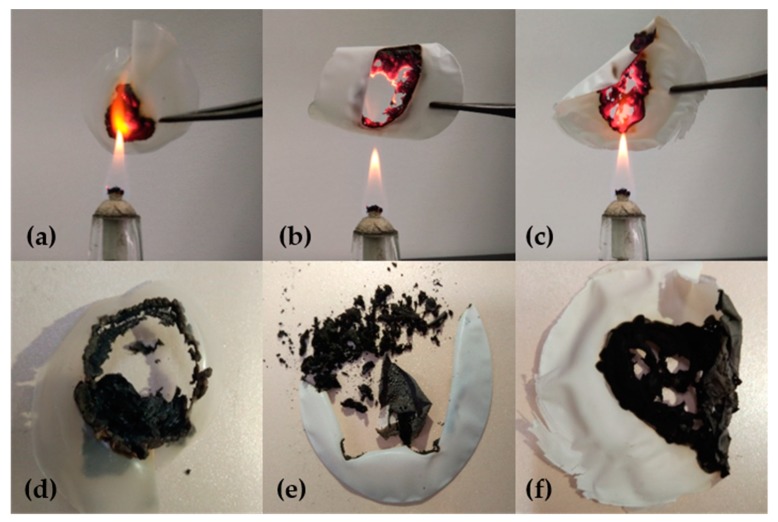
The photographs during and after the burning process of CMC (**a**,**d**), CMC/MAG–20 (**b**,**e**), CMC/MAG–40 (**c**,**f**).

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
