# Peer review of "Investigation on the Preparation and Properties of CMC/magadiite Nacre-Like Nanocomposite Films"

_polymers, 2019, doi:10.3390/polym11091378_

Round 1
Reviewer 1 Report
This is an interesting paper on the production of a composite of carboxymethylcellulose with magadiite mineral. The results are encouraging and worth of publication. However, the description of the results is upsetting, mostly because of the use of English. I am not a native speaker and my doubts are not of linquistic character. For exemple all the discussion given in lines 146-163 is very chaotic and thus difficult to follow. Also the description of the structure of the formed nanocomposite is seen well on the Figures, whereas its description is unclear. Thus paragraphs 3.1. and 3.2 require careful elaboration.
Author Response
Thanks for kind review of our paper. We have checked it carefully and explore more things as the comments by the reviewer. We have explained more clearly and more detail. English language has been revised and rechecked. I would like to request the reviewer to check the revised version. Thanks

Reviewer 2 Report
The present work involves synthesis and characterization of CMC/MAG composites with acceptable thermal and optical properties. Authors have done a fair job in characterizing the composites and have attempted to propose an intercalation mechanism based on SEM and XRD studies.
Few comments-
The word "research" is too broad and it can include just a literature review as well. Authors should consider using the terms- investigation, study, synthesis and characterization on the title Introduction has quiet a few abbreviations. Authors should spell out TG An elaborated mechanism on the formation of unique morphological structures is required to support the improved properties that are being demonstrated.
Author Response

(The authors gave the same response as above.)

Round 2
Reviewer 1 Report
After improvement paper is o.k. to me